# Activation of the P2RX7/IL-18 pathway in immune cells attenuates lung fibrosis

Serena Janho dit Hreich[1,2], Thierry Juhel[1], Sylvie Leroy[2,3,4], Alina Ghinet[5,6,7], Frederic Brau[3], Veronique Hofman[1,2,8,9], Paul Hofman[1,2,8,9], Valerie Vouret-Craviari[1,2]*

[1]Université Côte d'Azur, CNRS, INSERM, IRCAN, Nice, France; [2]FHU OncoAge, Nice, France; [3]Université Côte d'Azur, CNRS, Institut Pharmacologie Moléculaire et Cellulaire, Sophia-Antipolis, France; [4]Université Côte d'Azur, Centre Hospitalier Universitaire de Nice, Pneumology Department, Nice, France; [5]Inserm U995, LIRIC, Université de Lille, CHRU de Lille, Faculté de médecine – Pôle recherche, Place Verdun, Lille, France; [6]Hautes Etudes d'Ingénieur (HEI), JUNIA Hauts-de-France, UCLille, Laboratoire de chimie durable et santé, Lille, France; [7]'Al. I. Cuza' University of Iasi, Faculty of Chemistry, Iasi, Romania; [8]Laboratory of Clinical and Experimental Pathology and Biobank, Pasteur Hospital, Nice, France; [9]Hospital-Related Biobank (BB-0033-00025), Pasteur Hospital, Nice, France

*For correspondence:
valerie.vouret@univ-cotedazur.fr

**Abstract** Idiopathic pulmonary fibrosis (IPF) is an aggressive interstitial lung disease associated with progressive and irreversible deterioration of respiratory functions that lacks curative therapies. Despite IPF being associated with a dysregulated immune response, current antifibrotics aim only at limiting fibroproliferation. Transcriptomic analyses show that the *P2RX7/IL18/IFNG* axis is downregulated in IPF patients and that P2*RX7* has immunoregulatory functions. Using our positive modulator of P2*RX7*, we show that activation of the P2*RX7*/IL-18 axis in immune cells limits lung fibrosis progression in a mouse model by favoring an antifibrotic immune environment, with notably an enhanced IL-18-dependent IFN-γ production by lung T cells leading to a decreased production of IL-17 and TGFβ. Overall, we show the ability of the immune system to limit lung fibrosis progression by targeting the immunomodulator P2*RX7*. Hence, treatment with a small activator of P2*RX7* may represent a promising strategy to help patients with lung fibrosis.

## eLife assessment

This study presents a potentially **valuable** discovery which indicates that activation of the P2RX7 pathway by the small molecule HEI3090 can reduce lung fibrosis after its establishment by inflammatory damage. If confirmed, the study could clarify the role of specific immune networks in the establishment and progression of lung fibrosis. The presented data and analyses showing the efficacy of HEI3090 small molecule acting via the P2RX7 pathway in reducing lung fibrosis are **solid**. The studies also show that genetic deletion of P2RX7 itself can reduce the extent of fibrosis. P2RX7 can thus have distinct effects in various phases of the development of lung fibrosis. There is a need for additional definitive studies that specifically identify the discrete phases of when inflammasome activation via P2RX7 signaling can worsen fibrosis versus when the same signaling can be beneficial. It also needs to be established whether distinct immune cell populations mediate the detrimental and beneficial effects of P2RX7 activation in lung fibrosis.

## Introduction

Idiopathic pulmonary fibrosis (IPF) is an aggressive interstitial lung disease associated with progressive deterioration of respiratory function that is ultimately fatal. It is characterized by destruction of the lung architecture due to accumulation of fibroblasts and extracellular matrix proteins, resulting in increased lung stiffness and impaired normal breathing.

Pirfenidone and nintedanib have been FDA approved for the treatment of IPF since 2014. They target, respectively, the key fibrotic cytokine TGFβ and several receptor tyrosine kinases, thereby affecting fibroblast activation and extracellular matrix protein production (*Heukels et al., 2019*). However, they only slow down the progression of the disease, so new therapeutic strategies and targets are needed.

Fibrosis is also associated with inflammation. In fact, fibrosis is a process of excessive wound healing and tissue remodeling due to repeated epithelial injuries releasing damage-associated molecular patterns (DAMPs) that trigger both the adaptive and innate immune systems. Although inflammation has not been considered a target in IPF, due to unsuccessful initial clinical trials of anti-inflammatory drugs (Idiopathic *Raghu et al., 2012*) or cyclophosphamide during exacerbations (*Naccache et al., 2022*), growing evidence suggest that altering specific immune populations that promote or attenuate disease progression may be beneficial (*Shenderov et al., 2021*).

Extracellular adenosine triphosphate (eATP) is a DAMP, released in high concentrations from injured cells in IPF patients (*Riteau et al., 2010*). High levels of eATP are recognized by the P2X7 receptor (P2*RX*7) and P2*RX*7 participates in the establishment of the bleomycin (BLM) lung fibrosis mouse model (*Riteau et al., 2010*). Extracellular activation of P2*RX*7 by ATP (*Surprenant et al., 1996*) leads to conformational rearrangements allowing influx of $Ca^{2+}$ and $Na^+$ ions into and the efflux of $K^+$ ions from the cell (*Karasawa et al., 2017*) and to the formation of a large pore mediating macrophage cell death. The nature of this large pore was subjected to controverse until the recent discovery that the macropore function is intrinsic to P2*RX*7 and does not involve progressive pore dilatation (*Harkat et al., 2017*; *Pippel et al., 2017*). Another feature of P2*RX*7 is its ability to activate the NLRP3 inflammasome, an event linked to potassium efflux (*Di et al., 2018*; *Muñoz-Planillo et al., 2013*), which requires both P2*RX*7 and TWIK2 activation (*Di et al., 2018*) and leads to the release of mature IL-1β and IL-18 (*Perregaux et al., 2000*), through gasdermin D pores. Consequently, P2*RX*7 has the ability to trigger an immune response (*Janho Dit Hreich et al., 2023*).

IL-1β is a pro-inflammatory cytokine with high profibrotic properties, as it promotes collagen deposition through IL-17A and TGFβ production (*Doerner and Zuraw, 2009*; *Sutton et al., 2009*; *Wilson et al., 2010*) but also promotes the activation and recruitment of inflammatory cells such as eosinophils and neutrophils. Indeed, deficiency of IL-1βR or its blockade ameliorate experimental fibrosis (*Couillin et al., 2009*; *Gasse et al., 2009*; *Gasse et al., 2007*). In contrast, the role of IL-18 is not clear. Indeed, conflicting experimental studies show that IL-18 could either promote (*Zhang et al., 2019*) or attenuate (*Nakatani-Okuda et al., 2005*) fibrosis. However, high levels of IL-18BP, a natural antagonist of IL-18, are associated with reduced overall survival in IPF patients (*Nakanishi et al., 2021*), suggesting that the activity of IL-18 may be required for improved survival or alternatively that high levels of IL-18 are accompanied by more IL-18BP, which leads to poorer outcomes.

IL-18 was originally described as IFN-γ-inducing factor (IGIF) and therefore boosts IFN-γ production by T cells and natural killer (NK) cells (*Okamura et al., 1995*). Not only has IFN-γ antiproliferative properties (*Elias et al., 1987*) but it also inhibits TGFβ activity (*Ulloa et al., 1999*; *Varga et al., 1990*) and therefore inhibits fibroblast activation and differentiation into myofibroblasts, alleviates TGFβ-mediated immunosuppression, inhibits extracellular matrix accumulation and collagen production (*Duncan and Berman, 1985*; *Gillery et al., 1992*; *Gurujeyalakshmi and Giri, 1995*; *Yuan et al., 1999*) and thus promotes an antifibrotic immune microenvironment, making IFN-γ a cytokine with antifibrotic properties. However, parenteral systemic administration of IFN-γ failed in clinical trials (INSPIRE; NCT 00075998) (*King and Jobling, 2009*), whereas local administration by inhalation showed promising results (*Diaz et al., 2012*; *Fusiak et al., 2015*; *Prior and Haslam, 1992*; *Skaria et al., 2015*; *Smaldone, 2018*).

One way to increase IFN-γ production locally and selectively in the lung would be to alter the phenotype of T cells since the polarization of T lymphocytes has been shown to impact fibroblasts' fate and immune infiltrate (*Luzina et al., 2008*; *Shenderov et al., 2021*). Indeed, T cells have been recently shown to selectively kill myofibroblasts through IFN-γ release and limit the progression of

lung and liver fibrosis in preclinical models (*Sobecki et al., 2022*) and set up an immune memory in the long term since IFN-γ-producing tissue resident memory T cells protect against fibrosis progression (*Collins et al., 2016*), highlighting the importance IFN-γ-producing T cells in this disease. Accordingly, CD4+-producing IFN-γ T cells are decreased in IPF and correlate with a better prognosis in IPF patients (*Giri et al., 1986*; *Luzina et al., 2008*; *Xu et al., 2006*).

In light of IL-18's demonstrated capability to modulate the T-cell phenotype by eliciting IFN-γ, we have posited a novel therapeutic approach for pulmonary fibrosis. Our proposal involves augmenting local IFN-γ production through targeted intervention in the P2*RX*7/IL-18 axis. This strategy aims to harness the immunomodulatory potential of this axis, offering a promising avenue for therapeutic intervention in pulmonary fibrosis. To implement this strategy, we utilized HEI3090, a chemical compound developed in our laboratory. HEI3090 has been previously reported to enhance the channel activity and large pore opening of P2*RX*7 and also to increase IL-18 levels (but not IL-1β) upon activation of the P2*RX*7/NLRP3 pathway (*Douguet et al., 2021*).

## Results
### Expression of P2RX7 and IL-18 activity is downregulated in IPF patients

The canonical release of IL-18 is due to activation of the P2*RX*7/NLRP3 pathway (*Perregaux et al., 2000*). Since high levels of eATP are found in IPF patients (*Riteau et al., 2010*) and P2*RX*7 is activated by such levels, it was of particular interest to investigate the involvement of P2*RX*7 in this disease. We used a publicly available dataset of lung homogenates from control and IPF patients and compared the mRNA expression levels of P2*RX*7 and markers of fibrosis, namely *ACTA2*, *COL1A2*, *COL3A1*, and *TGFB3*. We found that the mRNA expression of P2*RX*7 is downregulated in IPF patients (*Figure 1A, B*), as well as the components of the NLRP3 inflammasome (*Figure 1—figure supplement 1*). Since IL-18 is constitutively expressed (*Puren et al., 1999*), which partly explains the lack of difference between control and IPF patients (*Figure 1B*), we investigated the signaling pathway downstream of IL-18. IL-18 binds to its receptor IL-18R1 coupled to its adaptor protein IL-18RAP which is required for IL-18 signaling and IFN-γ expression. We showed that *IL18R1*, *IL18RAP*, and *IFNG* (*Figure 1B*) are downregulated in IPF patients. Knowing that the modulation of the phenotype of T cells is promising in IPF (*Shenderov et al., 2021*), we checked whether *P2RX7* and *IL18* are linked to an immune response in IPF patients using Gene Set Enrichment Analyses. Indeed, we showed that the expression of *P2RX7* and *IL18* signaling (*IL18* and *IL18RAP*) correlates with the *IFNG* response as well as with immunoregulatory interactions required for changing the phenotype of T cells (*Figure 1C*, *Figure 1—figure supplement 1*). This transcriptomic analysis highlights that the *P2RX7/IL18* signaling pathway is downregulated and suggests that this pathway may be able to modulate the immune response in IPF patients.

### HEI3090 inhibits the onset of pulmonary fibrosis in the BLM mouse model

Even though transcriptomic changes do not always reflect changes in the proteome these encouraging results led us to hypothesize that activation of the P2*RX*7/IL-18 axis may restrain lung fibrosis progression by promoting an antifibrotic immune microenvironment. To test this hypothesis, we enhanced the P2*RX*7/IL-18 axis in the BLM-induced lung fibrosis mouse model by using HEI3090, a chemical compound described to stimulate immune cells expressing P2*RX*7 to generate IL-18 in the presence of eATP (*Douguet et al., 2021*).

We first tested the antifibrotic potential of HEI3090 on mice having an established fibrosis (*Figure 2A*). Activation of P2*RX*7 with HEI3090 in mice 7 days after BLM administration reduced the development of pulmonary fibrosis, as evidenced by less thickening of alveolar walls and free air space (*Figure 2B*). Fibrosis severity was evaluated using the Ashcroft score. To overcome the heterogeneity of fibrosis within the lobes, we scored the whole surface of the lung, and the result represents the mean of each field (*Figure 2C* and *Figure 2—figure supplement 1A, B*). This quantification showed that fibrosis is reduced by 40% in lungs of HEI-treated mice.

As accumulation of fibroblasts/myofibroblasts and extracellular matrix proteins are a hallmark of fibrosis, we also quantified fibroblasts/myofibroblasts and collagen levels in the lungs of vehicle

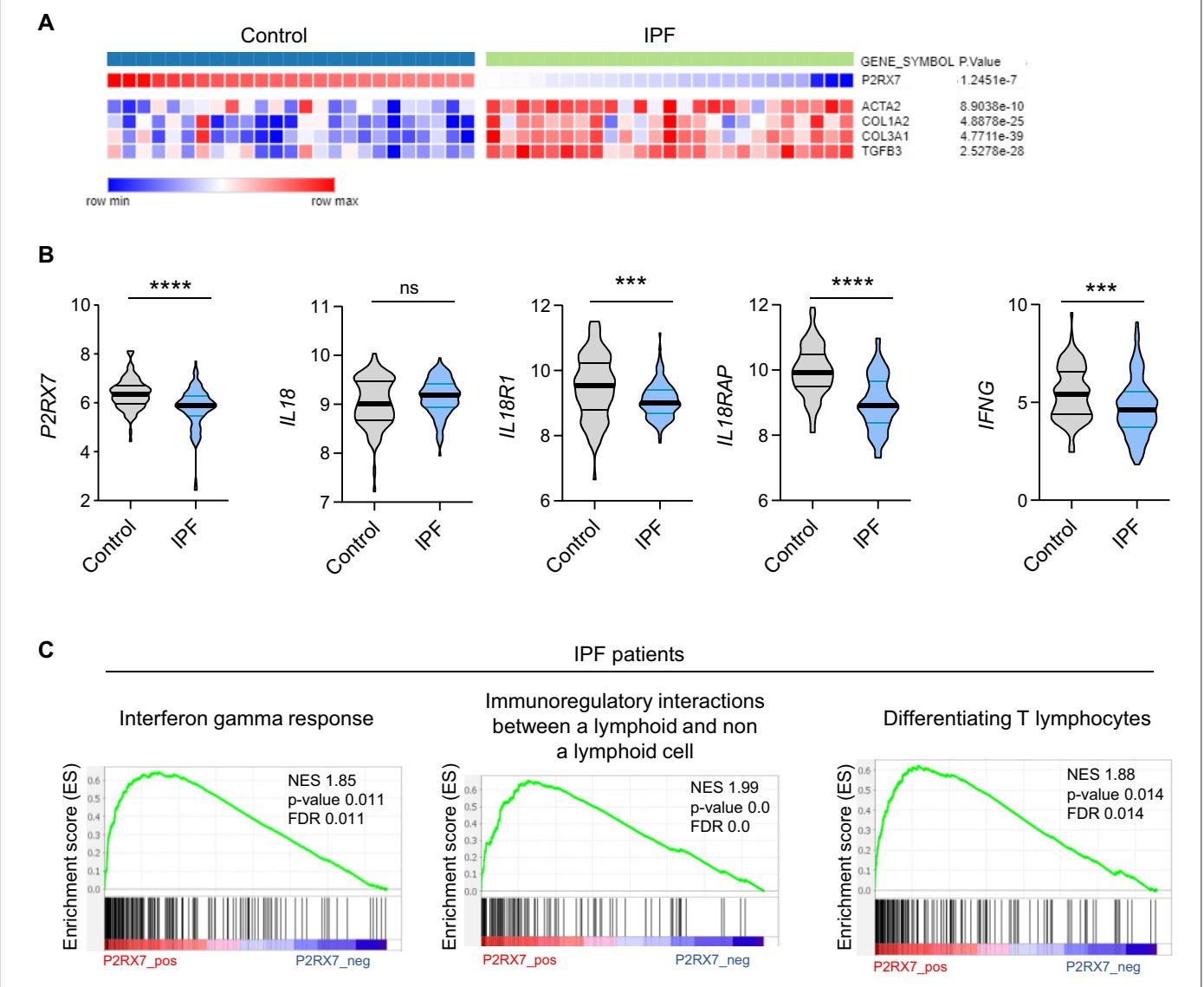

**Figure 1.** The P2*RX7*/IL-18/IFN-γ pathway is downregulated in idiopathic pulmonary fibrosis (IPF). (**A**) Heatmap of mRNA expression of P2*RX7* in control and IPF patients with a cluster of fibrosis-associated genes. Raw p-values are shown (Limma). p-values were determined by Spearman test. (**B**) mRNA expression of *P2RX7, IL18, IL18R1, IL18RAP*, and *IFNG* between control and IPF patients from 213 individuals, corresponding to 91 controls and 122 IPF patients. p-values were determined by two-tailed unpaired *t*-test or Mann–Whitney *t*-test. ***p < 0.001, ****p < 0.0001. (**C**) Gene set enrichment analysis (GSEA) plot associating P2*RX7* mRNA levels from IPF patients with three immunological signatures. The green line represents the enrichment score and the black lines the specific signature-associated genes. NES: normalized enrichment score. p-values (bilateral Kolmogorov–Smirnov) and false discovery rate (FDR) are shown.

The online version of this article includes the following source data and figure supplement(s) for figure 1:

**Source data 1.** Excel file containing output result of mRNA expression for the graph in *Figure 1B*.

**Figure supplement 1.** The components of the NLRP3 inflammasome are downregulated in idiopathic pulmonary fibrosis (IPF).

**Figure supplement 1—source data 1.** Excel file containing output result of mRNA expression for the graph in *Figure 1—figure supplement 1A*.

and HEI3090-treated mice using, respectively, an antibody which is specific to the PDGFRα and by measuring the intensity of Sirius Red-stained collagen fibers under polarized light. We showed that both the percentage of CD140a (PDGFRα)-positive cells in non-immune cells subset (*Figure 2D*) and the collagen fibers were reduced in lungs of HEI3090-treated mice (*Figure 2E, F*).

We also tested the ability of HEI3090 to limit lung fibrosis progression when administered during the inflammatory phase of the BLM model (*Figure 2G*) that mimics the exacerbation episodes in IPF

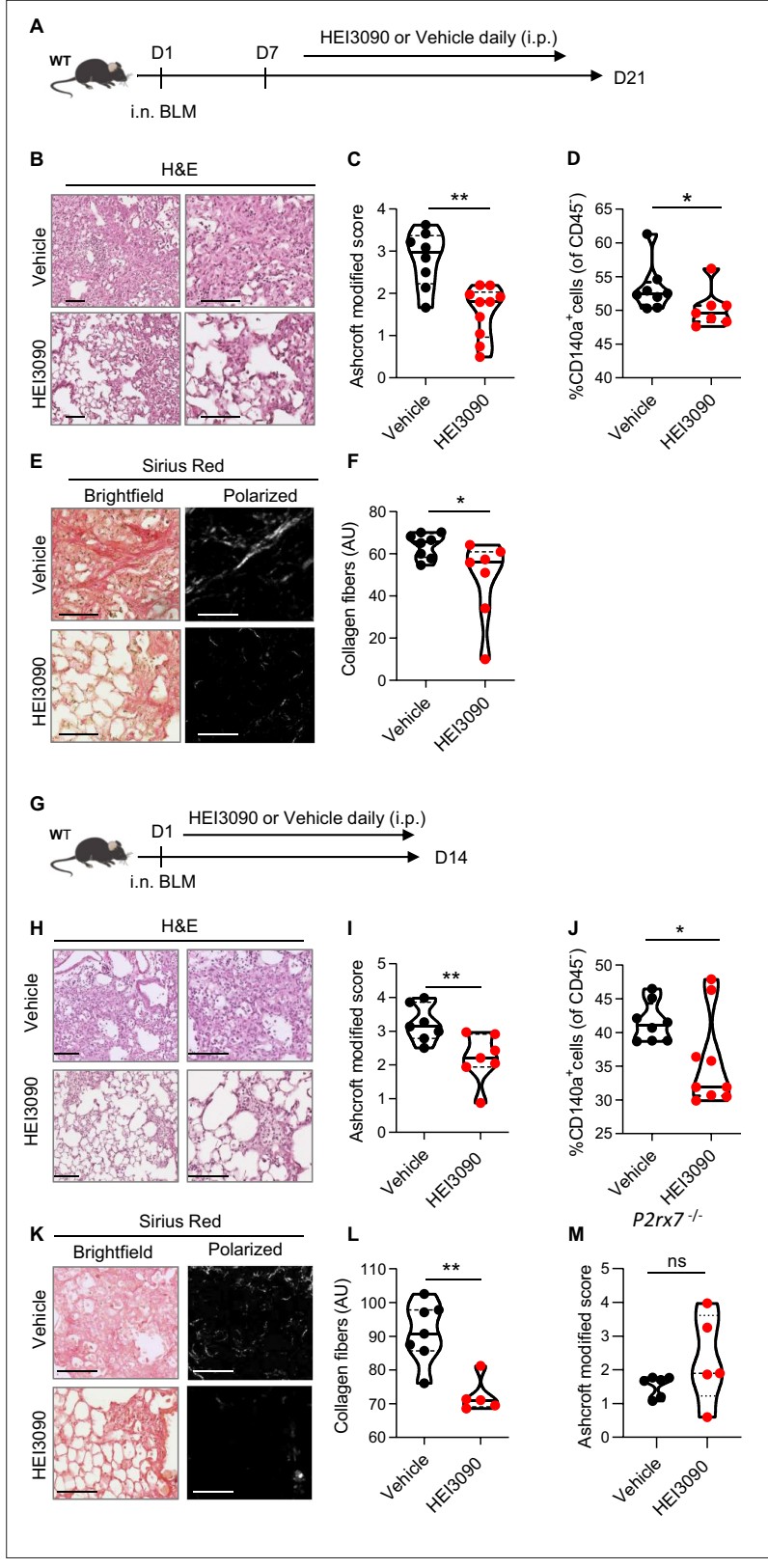

**Figure 2.** HEI3090 inhibits lung fibrosis progression. (**A**) Experimental design. WT mice were given 2.5 U/kg of bleomycin by i.n. route. At the end of the inflammatory phase (D7), 1.5 mg/kg of HEI3090 or vehicle were given daily until day 21. (**B**) Representative images of lung sections at day 21 after treatment stained with H&E and Sirius Red, scale bar = 100 μm. (**C**) Fibrosis score assessed by the Ashcroft method. p-values were determined by

*Figure 2 continued on next page*

*Figure 2 continued*

two-tailed unpaired *t*-test. (**D**) Quantification of fibroblasts/myofibroblasts in non-immune cell subset. p-values were determined by two-tailed Mann–Whitney test. (**E**) Representative images of lung sections at day 21 after treatment stained with Sirius Red, scale bar = 100 µm. (**F**) Collagen levels in whole lung of mice assessed on Sirius Red-polarized images. p-values were determined by two-tailed unpaired *t*-test. (**G**) Experimental design. WT mice were given 2.5 U/kg of bleomycin by i.n. route. 1.5 mg/kg of HEI3090 or vehicle were given daily until day 14. (**H**) Representative images of lung sections at day 14 after treatment stained with H&E, scale bar = 100 µm. (**I**) Fibrosis score assessed by the Ashcroft method. p-values were determined by two-tailed unpaired *t*-test. (**J**) Quantification of fibroblasts/myofibroblasts in non-immune cell subset. p-values were determined by two-tailed Mann–Whitney test. (**K**) Representative images of lung sections at day 14 after treatment stained with Sirius Red, scale bar = 100 µm. (**L**) Collagen levels in whole lung of mice assessed on Sirius Red-polarized images. p-values were determined by two-tailed Mann–Whitney test. (**M**) *P2rx7* KO mice were given 2.5 U/kg of bleomycin by i.n. route. 1.5 mg/kg of HEI3090 or vehicle were given daily until day 14. Fibrosis score assessed by the Ashcroft method is showed. p-values were determined by two-tailed Mann–Whitney test. For all analyses, the violin plot illustrates the distribution of Ashcroft scores across indicated groups. The width of the violin at each point represents the density of data, and the central line indicates the median expression level. Each point represents one biological replicate. *p < 0.05, **p < 0.01. WT: wildtype, BLM: bleomycin, i.p.: intraperitoneal, i.n.: intranasal, H&E: hematoxylin & eosin, AU: arbitrary units.

The online version of this article includes the following source data and figure supplement(s) for figure 2:

**Source data 1.** Excel file containing output results of fibrosis analysis for the graph in *Figure 2C, D, F, I, J, L, M*.

**Figure supplement 1.** HEI3090 treatment limits lung fibrosis progression.

**Figure supplement 1—source data 1.** Excel file containing output results of fibrosis analysis for the graph in *Figure 2—figure supplement 1C, D*.

patients (*Peng et al., 2013*). HEI3090 was also able to inhibit the onset of lung fibrosis in this setting given less damage on lung architecture (*Figure 2H*), the reduced fibrosis score (*Figure 2I*). We further evaluated the percentage of fibroblasts/myofibroblasts in non-immune cells subset (*Figure 2J*). At this shorter time point, we observed less fibroblasts/myofibroblasts in lungs of wildtype (WT) mice treated with vehicle and the number of CD140a-positive cells is decreased in lungs of HEI3090-treated mice. Further, and as for mice with an established fibrosis, HEI3090 reduced the intensity of collagen fibers in lungs of HEI3090-treated mice (*Figure 2K, L*). We further verified whether the antifibrotic action of HEI3090 depends on the expression of P2*RX*7 by inducing lung fibrosis in *P2rx7*−/− mice. In doing so, we initially observed that P2*RX*7 plays a role in the development of BLM-induced lung fibrosis. This is illustrated by a decrease of 50% in the Ashcroft score, with a mean value of 1.7 in P2*RX*7 knockout mice compared to 3 in WT mice (*Figure 2M* and *Figure 2—figure supplement 1C*). It is important to note that *P2rx7*−/− mice still exhibit signs of lung fibrosis, such as thickening of the alveolar wall and a reduction in free air space, in comparison to naïve mice that received phosphate-buffered saline (PBS) instead of BLM (see *Figure 2—figure supplement 1A*). This result confirms a previous report indicating that BLM-induced lung fibrosis partially depends on the activation of the P2*RX*7/pannexin-1 axis, leading to the production of IL-1β in the lung. Additionally, in contrast to the observations in WT mice, HEI3090 failed to attenuate the remaining lung fibrosis in *P2rx7*−/− mice, as measured by the Ashcroft score (*Figure 2M*), the percentage of lung tissue with fibrotic lesions, or the intensity of collagen fibers (*Figure 2—figure supplement 1D*). These results show that P2*RX*7 alone participates in fibrosis and that HEI3090 exerts a specific antifibrotic effect through this receptor (see *Figure 2—figure supplement 1C*).

## HEI3090 shapes immune cell infiltration in the lungs

Given that transcriptomic analysis revealed the immunoregulatory functions of *P2RX7* in IPF (*Figure 1*) and the antifibrotic activity of HEI3090 (*Figure 2*), we subsequently explored whether HEI3090 had an impact on both the immune landscape of the lung and the production of cytokines. This investigation aimed to elucidate the potential mechanisms underlying the dual antifibrotic effects observed with both P2*RX*7 deletion and P2*RX*7 activation by HEI3090.

We show that lung CD3+ T cells were more biased to produce IFN-γ than the profibrotic IL-17A cytokine when mice expressing P2*RX*7 were treated with HEI3090 (*Figure 3A* and *Figure 3—figure supplement 1A*). This biased production of IFN-γ is only seen in CD3+ T cells and not in overall lung

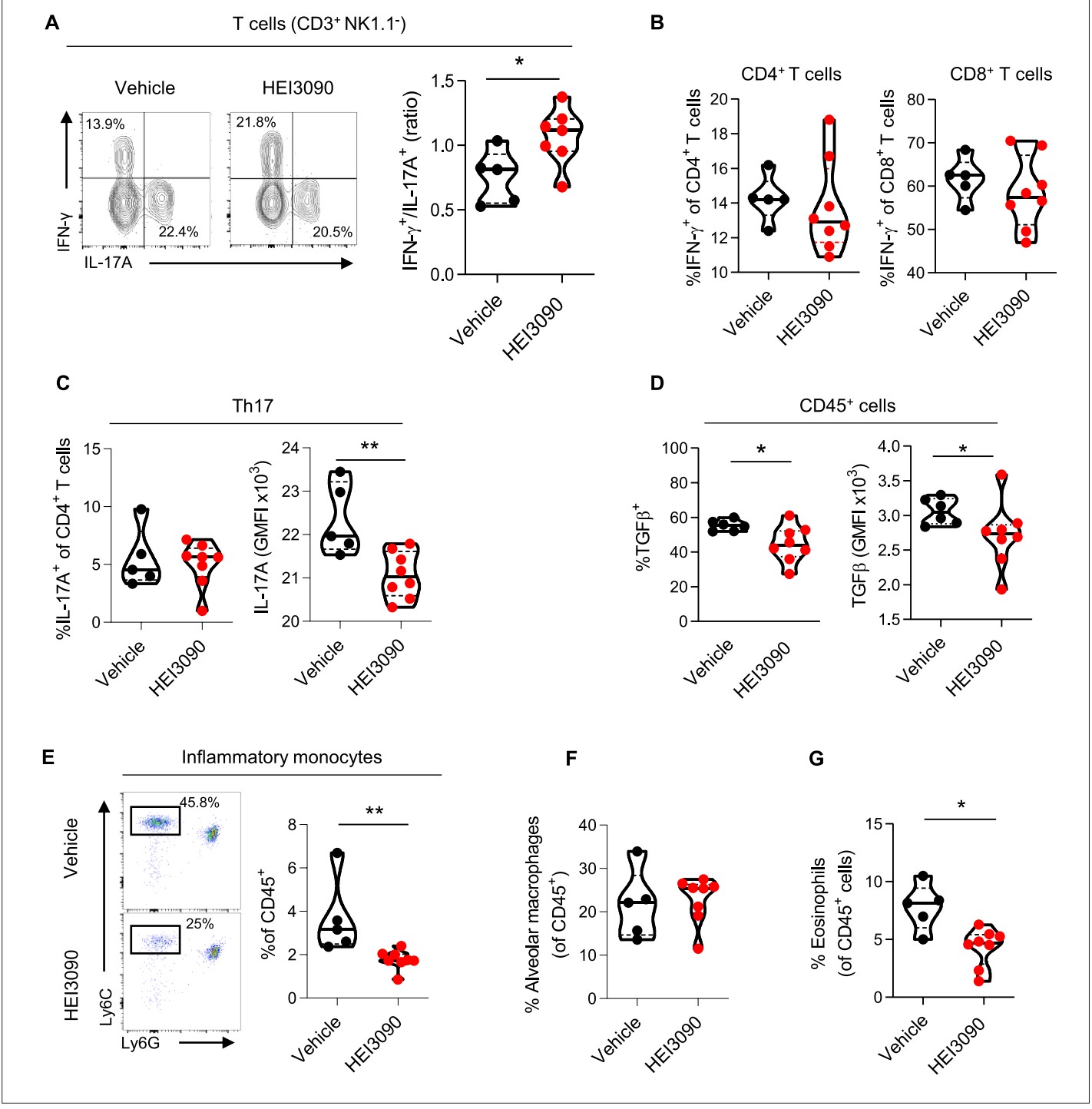

**Figure 3.** HEI3090 favors an antifibrotic immune signature in the lungs. Wildtype (WT) mice were given 2.5 U/kg of bleomycin by i.n. route and treated daily i.p. with 1.5 mg/kg of HEI3090 or Vehicle. Lungs were analyzed by flow cytometry at day 14. (**A**) Contour plot of IFN-γ- and IL-17A-producing T cells (CD3+NK1.1−) (left) and ratio of IFN-γ over IL-17A in T cells (CD3+NK1.1−) (right). p-values were determined by two-tailed Mann–Whitney test. (**B**) Percentage of IFN-γ-producing CD4+ and CD8+ T cells. p-values were determined by two-tailed Mann–Whitney test. (**C**) Percentage and GMFI of IL-17A+ cells of CD4+ T cells (CD3+CD4+NK1.1−). p-values were determined by two-tailed Mann–Whitney test. (**D**) Percentage and GMFI of TGFβ in CD45+ cells. p-values were determined by two-tailed Mann–Whitney test. (**E**) Dotplot showing lung inflammatory monocytes, gated on lineage−CD11c−CD11b+ cells (left) and percentage of lung inflammatory monocytes (Ly6Chigh Ly6G−) (right). p-values were determined by two-tailed Mann–Whitney test. (**F**) Percentage of alveolar macrophages (CD11c+SiglecF+), p-values were determined by two-tailed unpaired t-test and (**G**) lung eosinophils (CD11b+SiglecF+CD11c−), p-values were determined by two-tailed Mann–Whitney test. (**A–G**) The violin plot illustrates the distribution of Ashcroft scores across indicated groups.

*Figure 3 continued on next page*

*Figure 3 continued*

The width of the violin at each point represents the density of data, and the central line indicates the median expression level. Each point represents one biological replicate. *p<0.05, **p<0.01. GMFI: geometric mean fluorescence intensity, i.n.: intranasal, i.p.: intraperitoneal.

The online version of this article includes the following source data and figure supplement(s) for figure 3:

**Source data 1.** Excel file containing output results of FACS analysis for the graph in *Figure 3A–D, F, G*.

**Figure supplement 1.** Activation of P2*RX7* with HEI3090 reshapes immune infiltration in the lungs.

**Figure supplement 1—source data 1.** Excel file containing output results of FACS analysis for the graph in *Figure 3—figure supplement 1A–G*.

**Figure supplement 2.** HEI3090 reactivates an immune response.

**Figure supplement 2—source data 1.** Excel file containing output results of FACS analysis for the graph in *Figure 3—figure supplement 2B, D*.

immune cells (*Figure 3—figure supplement 1B*) nor in the subsets of T lymphocytes (*Figure 3B*, *Figure 3—figure supplement 1B*) or NK cells (*Figure 3—figure supplement 1C*). Although levels of CD3$^+$ T cells and T-cell subsets were unchanged (*Figure 3—figure supplement 1D*), including the profibrotic Th17 cells (*Figure 3C*), IL-17A production by Th17 cells is substantially attenuated after HEI3090 treatment (*Figure 3C*), consistent with the ability of IFN-γ to inhibit IL-17A production (*Laurence et al., 2007*). Considering the strong profibrotic properties of TGFβ and its mutual antagonism with IFN-γ (*Ishida et al., 2004*; *Park et al., 2007*), we checked whether HEI3090 had an effect on TGFβ levels. Indeed, treatment with HEI3090 reduced TGFβ-producing immune cells in the lung as well as TGFβ production (*Figure 3D*). Notably, HEI3090 treatment reduced TGFβ production in NK cells but not in T-cell subsets (*Figure 3—figure supplement 1F*).

Pulmonary fibrosis is also favored and driven by the recruitment of inflammatory cells, mainly from the myeloid lineage. Monocytes are highly inflammatory cells that are recruited to the lung and differentiate into alveolar macrophages, both of which have strong profibrotic properties (*Gibbons et al., 2011*; *McCubbrey et al., 2018*; *Misharin et al., 2017*). We demonstrated that in HEI3090-treated mice, the number of inflammatory monocytes decreased markedly (*Figure 3G*), whereas the number of alveolar macrophages remained unchanged (*Figure 3F*), consistent with the prognostic ability of monocyte count in IPF progression (*Achaiah et al., 2021*; *Kreuter et al., 2021*; *Liu et al., 2019*; *Scott et al., 2019*). We also examined other inflammatory cells with profibrotic properties, such as eosinophils (*Morimoto et al., 2018*) that are less present in HEI3090-treated lungs (*Figure 3G*), or PMN levels that remain unchanged by HEI3090 treatment (*Figure 3—figure supplement 1D*).

We also wondered if the activation of P2*RX7* has a systemic effect by analyzing immune changes in mice's spleens. No significant change in cell populations was observed in the spleens of mice (*Figure 3—figure supplement 2B*) when HEI3090 was administrated in the early phase of the BLM model suggesting a local lung activity of the molecule. However, HEI3090 reactivated a systemic immune response with higher levels of dendritic cells (DCs) and lymphocytes in the spleens of mice treated during the fibroproliferation phase (*Figure 3—figure supplement 2D*). These results show the ability of HEI3090 to shape the immune response locally and impact the progression of fibrosis systemically even in the fibroproliferative phase of the BLM model.

Altogether these results demonstrate that HEI3090 promotes an antifibrotic cytokinic profile in lung immune cells and attenuates lung inflammation.

## HEI3090 requires the P2RX7/NLRP3/IL-18 pathway in immune cells to inhibit lung fibrosis

We wanted to further investigate the mechanism of action of HEI3090 by identifying the cellular compartment and signaling pathway required for its activity. Since the expression of P2*RX7* and the P2*RX7*-dependent release of IL-18 are mostly associated with immune cells (*Ferrari et al., 2006*), and since HEI3090 shapes the lung immune landscape (*Figure 3*), we investigated whether immune cells were required for the antifibrotic effect of HEI3090. To do so, we conducted adoptive transfer experiments wherein immune cells from a donor mouse were intravenously injected 1 day before BLM administration into an acceptor mouse. The intravenous injection route was chosen as it is a standard method for targeting the lungs, as previously documented (*Wei and Zhao, 2014*). This approach was previously used with success in our laboratory (*Douguet et al., 2021*). It is noteworthy that this adoptive transfer approach did not influence the response to HEI3090. This was observed consistently in both *P2rx7*$^{-/-}$ mice and *P2rx7*$^{-/-}$ mice that received splenocytes of the same genetic background.

In both cases, HEI3090 failed to mitigate lung fibrosis, as depicted in *Figure 2M*, *Figure 2—figure supplement 1D*, and *Figure 4—figure supplement 2A, B*. First, we tested whether WT P2*RX*7-expressing splenocytes infuse into *P2rx7⁻/⁻* mice restored the antifibrotic activity of HEI3090. We show that restriction of P2*RX*7 expression on immune cells restored the antifibrotic effect of HEI3090 based on the architecture of the lung, with lungs of HEI3090-treated mice showing more free airspace and thinner alveolar walls (*Figure 4B*), as well as an overall lower fibrosis score (*Figure 4C*) and collagen fiber intensity (*Figure 4D*) than control lungs. Since it was previously showed that the BLM mouse model relies on P2*RX*7-expressing pulmonary epithelial cells (*Riteau et al., 2010*), we wanted to validate further the role of P2*RX*7-expressing immune cells in a mouse model where P2*RX*7 is expressed by non-immune cells. To do so, we reduced both the expression of P2*RX*7 and its activity by repeated i.v. administration (days 1, 4, 7 9, and 11) of *P2rx7⁻/⁻* splenocytes in WT mice (*Figure 4—figure supplement 1*). In this setting, HEI3090 was unable to limit the progression of fibrosis. These results highlight the important role of immune cells and rules out a major role of non-immune P2*RX*7-expressing cells, such as fibroblasts, in the antifibrotic effect of HEI3090.

To test the importance of the NLRP3/IL-18 pathway downstream of P2*RX*7, we performed an adoptive transfer of *Nlrp3⁻/⁻* and *Il18⁻/⁻* splenocytes into *P2rx7⁻/⁻* mice, expressing similar levels of P2*RX*7 as WT splenocytes (*Figure 4—figure supplement 2E*), but also the same levels of IL-18 and NLRP3 (*Figure 4—figure supplement 2F*). We also verified in this experimental setting by comparing each experimental group (treated and non-treated) that the genetic background did not affect lung fibrosis (*Figure 4—figure supplement 3*). The absence of NLRP3 and IL-18 in P2*RX*7-expressing immune cells abrogated the ability of HEI3090 to inhibit lung fibrosis because the lung architecture and collagen fibers intensity resembled that of control mice (*Figure 4E-J*). Consistent with the requirement of IL-18 for HEI3090's antifibrotic activity, activation of P2*RX*7 with this molecule in WT mice increased the levels of IL-18 in the sera of these mice compared to control mice (*Figure 4K*). Moreover, neutralization of IL-18 abrogated the increase of the IFN-γ/IL-17A ratio by lung T cells (*Figure 4L*), highlighting furthermore the necessity of IL-18 for the antifibrotic effect of HEI3090.

Not only does the activation of the P2*RX*7/NLRP3 pathway leads to the release of IL-18, but also induce the release of the pro-inflammatory and profibrotic IL-1β cytokine. However, IL-1β was not involved in the antifibrotic effect of HEI3090 (*Figure 4—figure supplement 2C*), nor were its levels affected by HEI3090 in WT mice (*Figure 4—figure supplement 2D*), in contrast to IL-18 levels measured in the same biological sample (*Figure 4K*).

Overall, we show that the P2*RX*7/NLRP3/IL-18 axis in immune cells is required to limit lung fibrosis progression, highlighting the efficacy in targeting the immune system in this disease.

## Discussion

A major unmet need in the field of IPF is new treatment to fight this uncurable disease. In this preclinical study, we demonstrate the ability of immune cells to limit lung fibrosis progression. Based on the hypothesis that a local activation of a T-cell immune response and upregulation of IFN-γ production has antifibrotic proprieties, we used the HEI3090-positive modulator of the purinergic receptor P2*RX*7, previously developed in our laboratory (*Douguet et al., 2021*), to demonstrate that activation of the P2*RX*7/IL-18 pathway attenuates lung fibrosis in the BLM mouse model. We have demonstrated that lung fibrosis progression is inhibited by HEI3090 in the fibrotic phase but also in the acute phase of the BLM fibrosis mouse model, that is during the period of inflammation. This lung fibrosis mouse model commonly employed in preclinical investigations, has recently been recognized as the optimal model for studying IPF (*Jenkins et al., 2017*). In this model, the intrapulmonary administration of BLM induces DNA damage in alveolar epithelial type 1 cells, triggering cellular demise and the release of ATP. The extracellular release of ATP from injured cells activates the P2*RX*7/pannexin 1 axis, initiating the maturation of IL-1β and subsequent induction of inflammation and fibrosis. In line with this, mice lacking P2*RX*7 exhibited reduced neutrophil counts in their bronchoalveolar fluids and decreased levels of IL-1β in their lungs compared to WT mice (*Riteau et al., 2010*). Based on these findings, Riteau et al. postulated that the inhibition of P2*RX*7 activity may offer a potential strategy for the therapeutic control of fibrosis in lung injury. In the present study, we provided strong evidence showing that selective activation of P2*RX*7 on immune cells, through the use of HEI3090, can dampen inflammation and fibrosis by releasing IL-18. The efficacy of HEI3090 to inhibit lung fibrosis was evaluated histologically on the whole lung's surface by evaluating the severity of fibrosis using three

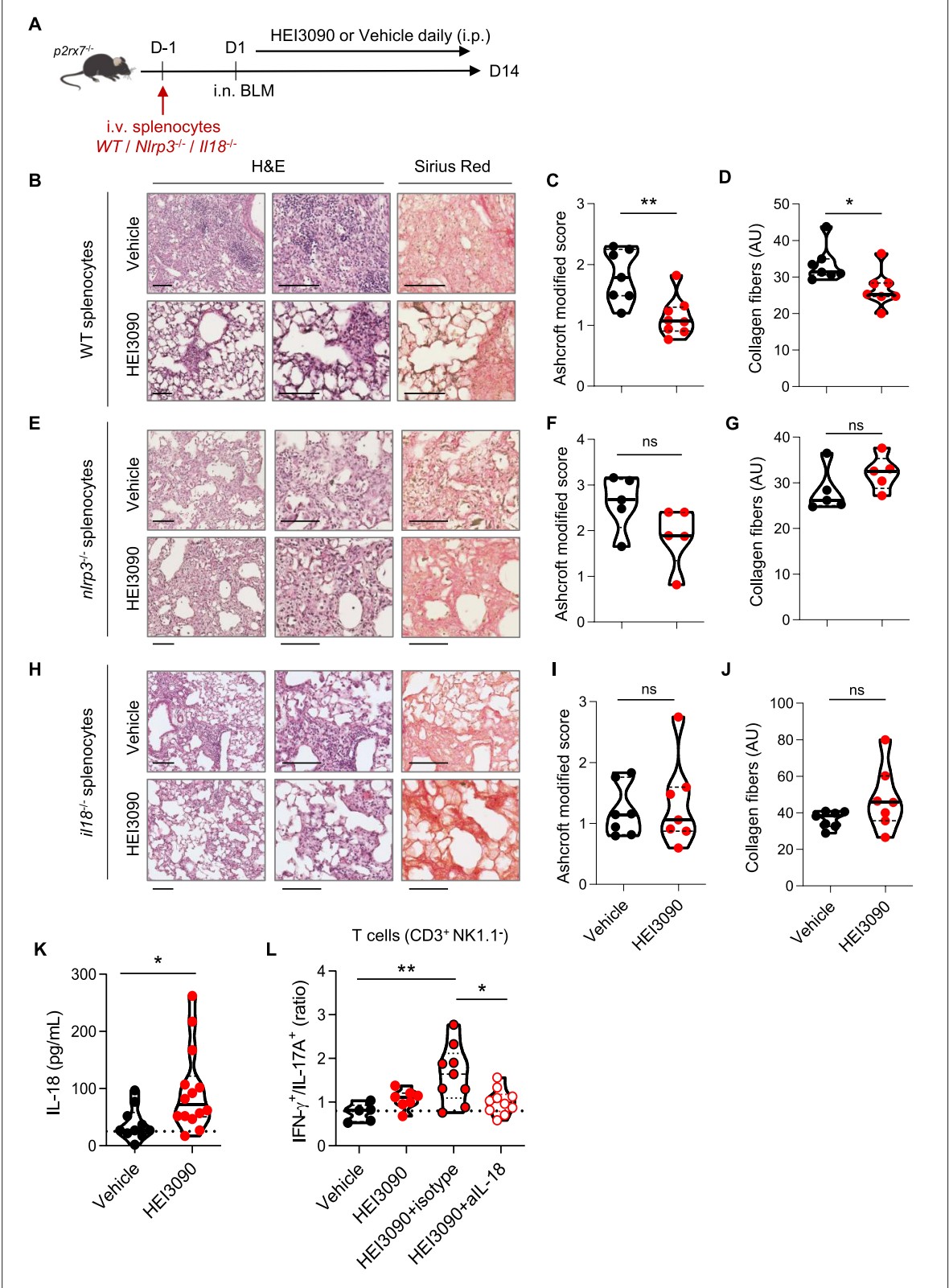

**Figure 4.** The P2RX7/NLRP3/IL-18 pathway in immune cells is required for HEI3090's antifibrotic effect. (**A**) Experimental design. $P2rx7^{-/-}$ mice were given 3.106 WT, $Nlrp3^{-/-}$, or $Il18^{-/-}$ splenocytes i.v. 1 day prior to BLM delivery (i.n. 2.5 U/kg). Mice were treated daily i.p. with 1.5 mg/kg HEI3090 or vehicle for 14 days. (**B, E, H**) Representative images of lung sections at day 14 after treatment stained with H&E and Sirius Red, scale bar = 100 µm. (**C, F, I**), Fibrosis score assessed by the Ashcroft method and (**D, G, J**) collagen fiber intensity. (**C**) WT splenocytes, p-values were determined by two-tailed

*Figure 4 continued on next page*

*Figure 4 continued*

Mann–Whitney test. (**D**) WT splenocytes, p-values were determined by two-tailed unpaired *t*-test. (**F, G**) *Nlrp3⁻/⁻* splenocytes, p-values were determined by two-tailed Mann–Whitney test. (**I, J**) *Il18⁻/⁻* splenocytes, p-values were determined by two-tailed Mann–Whitney test. (**K**) IL-18 levels in sera of WT BLM-induced mice at day 14. p-values were determined by two-tailed unpaired *t*-test. (**L**) Ratio of IFN-γ over IL-17A in lung T cells (CD3⁺NK1.1⁻) of WT mice neutralized for IL-18 or not (isotype control) every 3 days. p-values were determined by one-way analysis of variance (ANOVA) with Tukey's multiple comparisons test. For all analyses, the violin plot illustrates the distribution of Ashcroft scores across indicated groups. The width of the violin at each point represents the density of data, and the central line indicates the median expression level. Each point represents one biological replicate. *p < 0.05, **p < 0.01. WT: wildtype, BLM: bleomycin, i.p.: intraperitoneal, i.n.: intranasal, i.v.: intravenous, H&E: hematoxylin & eosin.

The online version of this article includes the following source data and figure supplement(s) for figure 4:

**Source data 1.** Excel file containing output results of analysis for the graph in *Figure 4B, D, F, G, I–L*.

**Figure supplement 1.** HEI3090 activity requires P2*RX7*'s expressing immune cells.

**Figure supplement 1—source data 1.** Excel file containing output results of Ashcroft score analysis for the graph in *Figure 4—figure supplement 1C*.

**Figure supplement 2.** The antifibrotic effect of HEI3090 is independent of IL1B expression.

**Figure supplement 2—source data 1.** Excel file containing output results analysis for the graph in *Figure 4—figure supplement 2B–E*.

**Figure supplement 2—source data 2.** Original file for the Western blot analysis in *Figure 4—figure supplement 2F* (anti-ACTB).

**Figure supplement 2—source data 3.** Original file for the Western blot analysis in *Figure 4—figure supplement 2F* (anti-IL-18).

**Figure supplement 2—source data 4.** Original file for the Western blot analysis in *Figure 4—figure supplement 2F* (anti-NLRP3).

**Figure supplement 2—source data 5.** PDF containing *Figure 4—figure supplement 2F* and original scans of the relevant Western blot analysis (anti-NLRP3, anti-IL-18, and anti-ACTB) with highlighted bands and sample labels.

**Figure supplement 3.** The genetic background does not impact lung fibrosis at steady step levels.

**Figure supplement 3—source data 1.** Excel file containing output results of Ashcroft score analysis for the graph in *Figure 4—figure supplement 3*.

independent approaches applied to the whole lung, the Ashcroft score, quantification of fibroblasts/myofibroblasts (CD140a) and polarized light microscopy of Sirius Red staining to quantify collagen fibers. All these methods of fibrosis assessment revealed that HEI3090 exerts an inhibitory effect on lung fibrosis, underscoring the necessity for a thorough preclinical assessment of HEI3090's mode of action. Notably, HEI3090 functions as an activator, rather than an inhibitor, of P2*RX7*, further emphasizing the importance of elucidating its intricate mechanisms.

Our study showed that inhibition of fibrosis by HEI3090 was associated with an increased production of IFN-γ by lung T cells that was dependent on an increased release of IL-18. The transcriptional analysis additionally indicated the downregulation of the P2*RX7*/IL-18/IFN-γ pathway in patients with IPF, where TGFβ levels are high (*Khalil et al., 1991*). This is consistent with the ability of TGFβ to downregulate IL-18R expression and IL-18-mediated IFN-γ production (*Koutoulaki et al., 2010*). These results confirm the beneficial effects of enhancing activation of the P2*RX7*/IL-18/IFN-γ pathway.

P2*RX7* is expressed by various immune and non-immune cells, but its expression is the highest in DCs and macrophages (*Di Virgilio and Vuerich, 2015*), from which IL-18 is mainly released to shape the T-cell response (*Sáez et al., 2017*) and increase T-cell IFN-γ production (*Iwai et al., 2008*). Collectively, and consistent with the immunomodulatory properties of P2*RX7*, these observations suggest that HEI3090 may target P2*RX7*-expressing antigen-presenting cells to influence the T-cell response, which may explain the selective T-cell increase in IFN-γ in HEI3090-treated mice. Accordingly, we have previously shown that HEI3090 targets the P2*RX7*/IL-18 axis in DCs to shape the immune response in a lung tumor mouse model (*Douguet et al., 2021*). Noteworthy, in contrast to parenteral administration of IFN-γ which failed in clinical trial (*King and Jobling, 2009*), we did not observe any sign of systemic side effect likely because HEI3090 needs ATP to potentiate P2*RX7* and ATP is present only in injured tissues.

HEI3090 not only increased IFN-γ production by T cells but it also reshaped the immune and cytokinic composition of the lung. Indeed, lungs of HEI3090-treated mice show a decrease in IL-17A production by Th17 cells and TGFβ production by lung immune cells. Moreover, lung inflammation is dampened after HEI3090 treatment, since the number of inflammatory monocytes and eosinophils decreases. It is not known whether this cytokinic and anti-inflammatory switch is solely due to the IL-17A and TGFβ-suppressive property of IFN-γ (*de Bruin et al., 2010*; *Iwamoto et al., 1993*; *Penton-Rol et al., 1998*), or whether it is a combination with the cell death-inducing property of P2*RX7* (*Surprenant et al., 1996*).

Of note, P2RX7 being known to be pro-inflammatory, through IL-1β release, our strategy to activate P2RX7 with HEI3090 may be considered as risky. However, in line with our prior in vivo and in vitro investigations (**Douguet et al., 2021**), HEI3090 failed to augment the release of IL-1β in the serum of mice treated with BLM, despite effectively increasing the release of IL-18. This observation elucidates why HEI3090 is effective in alleviating lung fibrosis. In agreement, previous reports have indicated that ATP stimulation of alveolar macrophages, derived from both IPF and lung cancer patients, led exclusively to an elevation in IL-18 levels (**Lasithiotaki et al., 2018**; **Lasithiotaki et al., 2016**), which was explained by an impaired NLRP3 inflammasome and a defective autophagy in IPF patients (**Patel et al., 2012**). Autophagy can be regulated by P2RX7 (**de Bruin et al., 2010**). Indeed, whereas basal autophagy suppresses NLRP3-inflammasome activation and cytokine secretion, induced autophagy augments IL-1β secretion (**Dupont et al., 2011**), therefore ruling out the hypothesis of HEI3090-induced autophagy to control the production of IL-1β. Unlike IL-1β, IL-18 is constitutively expressed in human and mouse immune cells (**Puren et al., 1999**) but also in non-immune cells such as fibroblasts (**Artlett et al., 2011**) and lung epithelium (**Watanabe et al., 2009**) and is both matured and released after NLRP3 activation. Therefore, it is currently not known whether the lack of IL-1β release is due to the different cytokine expression pattern or whether there is IL-1β-specific regulation following an enhanced activation of P2RX7 and additional studies are required to answer this question.

In addition to HEI3090, other molecules have been suggested to be positive modulators of P2RX7 in vitro (**Stokes et al., 2020**). Among them the most described are clemastine and ginsenosides. Clemastine, a first generation anti-histaminic, enhances ATP-induced P2RX7-dependent Ca²⁺ entry, macropore opening and IL1B production in immune cells (**Nörenberg et al., 2011**). However, when tested in vivo clemastine treatment does not activates P2RX7, indeed it reduces the production of pro inflammatory cytokines and decreases P2RX7 expression in hippocampus (**Su et al., 2018**). These biological responses correspond to an inhibitory effect on P2RX7. Ginsenosides, purified from root of Panax ginseng, have been shown to potentiate ATP-induced P2RX7-dependent Ca²⁺ entry and macropore opening but also cell death in vitro (**Helliwell et al., 2015**). So far, there is no evidence indicating that ginsenosides act as positive modulators of P2RX7 in vivo and all the other chemical compounds described to be active in vivo are negative allosteric modulator of P2RX7 (**Karasawa and Kawate, 2016**).

The novelty of our approach is that it targets and alters the immune environment of the lung. Indeed, the use of a P2RX7-specific modulator that acts only in an ATP-rich environment was effective in promoting an anti-inflammatory and antifibrotic phenotype by altering several key mediators of the disease. Therapies given to IPF patients are known to induce significant side effects which can limit their prescription (**Raghu et al., 2015**). Since current therapies (pirfenidone or nintedanib) and HEI3090 have different mechanisms of action, it could be interesting to test in preclinical studies the efficacy of a treatment combining lower doses of current molecules and HEI3090. If successful, this may open a new therapeutic option to be tested in IPF patients.

In this study, we emphasize the importance of IL-18 for an antifibrotic effect. Several studies have indicated that P2RX7/NLRP3/IL-18 promote disease progression using knock out mice or inhibitors (**Artlett et al., 2011**; **Riteau et al., 2010**; **Zhang et al., 2019**). However, experimental mouse models rely on lung epithelial cell injury that has been shown to activate the NLRP3 inflammasome in the lung epithelium as a first step (**Gasse et al., 2009**; **Riteau et al., 2010**) and release danger signals that activate the immune system as a second step. Therefore, initial lung injury to epithelial cells is reduced or absent in *p2rx7⁻/⁻* and *nlrp3⁻/⁻* mice, indicating that P2RX7 and NLRP3 are required for the establishment of the BLM mouse model rather than their role in an already established fibrosis, which has not yet been studied. In addition, NLRP3 and the release of IL-1β and IL-18 from fibroblasts have been shown to promote myofibroblast differentiation and extracellular matrix production (**Artlett et al., 2011**; **Pinar et al., 2020**; **Watanabe et al., 2009**). These observations suggest that fibrosis mouse models initially rely on NLRP3 activation by non-immune cells and encourage further studies on the contribution of the NLRP3 inflammasome in immune cells to fibrosis progression in vivo. Since we have highlighted the importance of this pathway in immune cells for delaying fibrosis progression, we propose that IL-18 may have different effects depending on the cell type. This hypothesis could be substantiated through experimentation utilizing inducible NLRP3 and P2RX7 knockout mice, wherein the invalidation of NLRP3 and P2RX7 would be induced after the establishment of fibrosis.

Beside an urgent need of new treatments for IPF, there is also a lack of biomarkers, such as prognostic biomarkers, markers of disease activity, or markers of drug efficacy. Our results suggest the possible benefit of an active IL-18 in the pathophysiology of pulmonary fibrosis and warrant analysis of IL-18 as a promising biomarker for predicting outcome in IPF patients. Given the potential effects of pirfenidone and nintedanib on IL-18 levels in preclinical models (*Mavrogiannis et al., 2022*; *Oku et al., 2008*; *Sharawy and Serrya, 2020*), determining IL-18 shifts during treatment would be highly interesting to evaluate potential changes in patients' outcome and to examine IL-18 levels which may be helpful in the long run for patient treatment strategy and subsequent introduction of pipeline drugs (*Spagnolo et al., 2021*).

Overall, we highlight in this study the ability of the P2*RX*7/NLRP3/IL-18 pathway in immune cells to inhibit the onset of lung fibrosis by using a positive modulator of P2*RX*7 that acts selectively in an eATP-rich environment such as fibrotic lung. The unique feature of this strategy is that it enhances the antifibrotic and it attenuates the profibrotic properties of immune cells, with no reported side effects, at least in mice.

# Methods

## Key resources table

| Reagent type (species) or resource | Designation | Source or reference | Identifiers | Additional information |
|---|---|---|---|---|
| Strain, strain background (*Mus musculus*) | C57BL/6 OlaHsD | Envigo | | WT mouse, male |
| Strain, strain background (*Mus musculus*) | *P2rx7*$^{-/-}$ | Jackson Laboratory | B6.129P2-P2rx7tm1Gab/J | Transgenic mouse, male |
| Antibody | anti-CD16/CD32 (Mouse monoclonal) | BD Bioscience | Clone 2.4G2, #553142 | FACS (1/100) |
| Antibody | anti-CD3A (Mouse monoclonal) | BD Bioscience | Clone 145-2611 #551163 | FACS (1/100) PerCP-Cy5.5 |
| Antibody | Anti NK1.1 (Mouse monoclonal) | BD Bioscience | Clone PK136 #562864 | FACS (1/100) PE-CF594 |
| Antibody | CD11c (Mouse monoclonal) | BD Bioscience | Clone HL3 #558079 | FACS (1/100) PE-Cy7 |
| Antibody | Ly6G (Mouse monoclonal) | BD Bioscience | Clone 1A8 #561236 | FACS (1/100) AF700 |
| Antibody | Ly6C (Mouse monoclonal) | BD Bioscience | Clone AL-21 #560594 | FACS (1/100) V450 |
| Antibody | SiglecF (Mouse monoclonal) | BD Bioscience | Clone E50-2440 #740388 | FACS (1/50) BV605 |
| Antibody | CD4 (Mouse monoclonal) | BD Bioscience | Clone RM4-5 #563726 | FACS (1/100) BV711 |
| Antibody | CD8α (Mouse monoclonal) | BD Bioscience | Clone 53-6.7 #100741 | FACS (1/100) BV650 |
| Antibody | Ly6G (Mouse monoclonal) | BD Bioscience | Clone 1A8 #561236 | FACS (1/100) AF700 |
| Antibody | Ly6C (Mouse monoclonal) | BD Bioscience | Clone AL-21 #560594 | FACS (1/100) V450 |
| Antibody | SiglecF (Mouse monoclonal) | BD Bioscience | Clone E50-2440 #740388 | FACS (1/50) BV605 |
| Antibody | CD4 (Mouse monoclonal) | BD Bioscience | Clone RM4-5 #563726 | FACS (1/100) BV711 |
| Antibody | CD8α (Mouse monoclonal) | BD Bioscience | Clone 53-6.7 #100741 | FACS (1/100) BV650 |

*Continued on next page*

*Continued*

| Reagent type (species) or resource | Designation | Source or reference | Identifiers | Additional information |
|---|---|---|---|---|
| Antibody | CD45.2 (Mouse monoclonal) | BD Bioscience | Clone 104 #563686 | FACS (1/100) BV786 |
| Antibody | Ly6G (Mouse monoclonal) | BD Bioscience | Clone 1A8 #561236 | FACS (1/100) AF700 |
| Antibody | Ly6C (Mouse monoclonal) | BD Bioscience | Clone AL-21 #560594 | FACS (1/100) V450 |
| Antibody | SiglecF (Mouse monoclonal) | BD Bioscience | Clone E50-2440 #740388 | FACS (1/50) BV605 |
| Antibody | CD4 (Mouse monoclonal) | BD Bioscience | Clone RM4-5 #563726 | FACS (1/100) BV711 |
| Antibody | CD8α (Mouse monoclonal) | BD Bioscience | Clone 53-6.7 #100741 | FACS (1/100) BV650 |
| Antibody | CD45.2 (Mouse monoclonal) | BD Bioscience | Clone 104 #563686 | FACS (1/100) BV786 |
| Antibody | CD45 (Mouse monoclonal) | BD Bioscience | Clone 30-F11 #564279 | FACS (1/100) BUV395 |
| Antibody | γδ TCR (Mouse monoclonal) | BD Bioscience | Clone GL3 #553178 | FACS (1/100) PE |
| Antibody | IFN-γ (Mouse monoclonal) | BD Bioscience | Clone XMG1.2 #554413 | FACS (1/100) APC |
| Antibody | IL-17A (Mouse monoclonal) | BD Bioscience | Clone TC11-18H10 #561718 | FACS (1/100) AF700 |
| Antibody | I-A/I-E (Mouse monoclonal) | Biolegend | Clone M5/114.15.2 #107651 | FACS (1/100) APCfire750 |
| Antibody | CD64 (Mouse monoclonal) | Biolegend | Clone X54-5/7.1 #139303 | FACS (1/20) PE |
| Antibody | P2RX7 (Mouse monoclonal) | Biolegend | Clone 1F11 #148706 | FACS (1/8) PE |
| Antibody | CD140a (Mouse monoclonal) | Biolegend | Clone APA5 #135905 | FACS (1/100) PE |
| Antibody | CD11b (Mouse monoclonal) | eBiosciences | Clone M1/70 #17-0112-82 | FACS (1/400) APC |
| Antibody | Foxp3 (Mouse monoclonal) | eBiosciences | Clone FJK16S #12-5773-82 | FACS (1/100) PE |
| Antibody | NLRP3 (Mouse monoclonal) | Adipogen | Clone Cryo-2 # AG-20B-0014 | WB (1/1000) |
| Antibody | ACTIN BETA (Mouse monoclonal) | Bio-Rad | #VMA00048 | WB (1/60000) |
| Antibody | IL-18 (Rabbit Polyclonal) | Biovision | #5180R-100 | WB (1/250) |
| Antibody | IL-18 (Mouse monoclonal) | BioXcell | Clone YIGIF74-1G7 #BE0237 | Neutralizing antibody (200 uμg per injection) |
| Commercial assay | Mouse IL-1 beta/IL-1F2 | R&D | #DY401 | ELISA |
| Commercial assay | Mouse IL-18 | MBL | #7625 | ELISA |
| Software | Prism8 | https://www.graphpad.com RRID:SCR_002797 | | |

*Continued on next page*

*Continued*

| Reagent type (species) or resource | Designation | Source or reference | Identifiers | Additional information |
|---|---|---|---|---|
| Software | FlowJo v10 | https://www.flowjo.com<br>RRID:SCR_008520 | | |
| Software | ImageJ 2.1.0 | https://imageJ.net/ImageJ<br>RRID:SCE_002285 | | |
| Chemical compound | Bleomycin Sulfate | Sigma-Aldrich | #B5507 | (2.5 U/kg) |
| Chemical compound | Ketamine | VIRBAC | | Anesthesia (25 mg/kg) |
| Chemical compound | Xylasine | Dechra | Sedaxylan | Anesthesia (2.5 mg/k) |
| Chemical compound | Phorbol 12- myristate 13-acetate | Sigma-Aldrich | # P1585 | 50 ng/ml |
| Chemical compound | Ionomycin | Sigma-Aldrich | # IO634 | 0.5 µg/ml |
| Commercial kit | Golgi Plug | BD Biosciences | # 555028 | 1 µl/ml |
| Commercial kit | Cytofix/Cytoperm | BD Biosciences | # 554722 | |
| Commercial kit | ACK lysis buffer | Gibco | A1049201 | |
| Other | Live Dead | Invitrogen | # L23102 | Use to stain dead cells |
| Commercial kit | Lung dissociation kit | Miltenyi Biotech | #130-095-927 | |
| Commercial kit | Spleen dissociation kit | Miltenyi Biotech | #130-095-928 | |
| Other | HEI3090 | | | PAM of P2*RX7*, **Douguet et al., 2021** |
| Other | H&E staining | Abcam | #ab245880 | Used for histology |
| Other | Picro Sirius Red | Abcam | # ab150681 | Used for histology |

## Data availability

In silico data used in this study were retrieved from Gene Expression Omnibus (GEO). We used the Gene expression profiling of 'chronic lung disease for the Lung Genomics Research Consortium cohort' (GSE47460, GPL14550). 122 patients with usual interstitial pneumonitis/IPF and 91 healthy controls were analyzed in this study. Microarray was done on whole lung homogenate from these subjects. Expression profile belong to the Lung Tissue Research Consortium (LTRC).

## Mice

Mice were housed under standardized light–dark cycles in a temperature-controlled air-conditioned environment under specific pathogen-free conditions at IRCAN, Nice, France, with free access to food and water. All mouse studies were approved by the committee for Research and Ethics of the local authorities (CIEPAL #598, protocol number APAFIS 21052-2019060610506376) and followed the European directive 2010/63/UE, in agreement with the ARRIVE guidelines. Experiments were performed in accord with animal protection representative at IRCAN. *P2rx7*$^{-/-}$ (B6.129P2-P2*rx*7tm1Gab/J, from the Jackson Laboratory) were backcrossed with C57BL/6J OlaHsD mice. C57BL/6J OlaHsD male mice (WT) were supplied from Envigo (Gannat, France). The determination of experimental group sizes involved conducting a pilot experiment with four mice in each group. Subsequently, a power analysis, based on the pilot experiment's findings (which revealed a 40% difference with a standard error of 0.9, $\alpha$ risk of 0.05, and power of 0.8), was performed to ascertain the appropriate group size for studying the effects of HEI3090 on BLM-induced lung fibrosis. The results of the pilot experiment and power analysis indicated that a group size of four mice was sufficient to characterize the observed effects. For each full-scale experiment, we initiated the study with 6–8 mice per group, ensuring a minimum of 5 mice in each group for robust statistical analysis. Additionally, we systematically employed the

ROULT method to identify and subsequently exclude any outliers present in each experiment before conducting statistical analyses.

## Induction of lung fibrosis

WT or $P2rx7^{-/-}$ male mice (8 weeks) were anesthetized with ketamine (25 mg/kg) and xylazine (2.5 mg/kg) under light isoflurane and were given 2.5 U/kg of BLM sulfate (Sigma-Aldrich) by intranasal route. Mice were treated i.p. every day with vehicle (PBS, 10% Dimethyl sulfoxide [DMSO]) or with HEI3090 (1.5 mg/kg in PBS, 10% DMSO) starting D1 or D7 post BLM delivery, as mentioned in the figures. After 14 days of treatment, lungs were either fixed for paraffin embedding or weighted and analyzed by flow cytometry. When mentioned, 200 µg of anti-IL-18 neutralizing antibody (BioXcell) or isotype control (IgG2a, BioXcell) were given by i.p. every 3 days starting 1 day prior to BLM administration.

## Adoptive transfer in *p2rx7*-deficient mice

Spleens from C57BL/6J male mice (8–10 weeks) were collected and digested with the spleen dissociation kit (Miltenyi Biotech) according to the supplier's instructions. $3 \times 10^6$ splenocytes were injected i.v. in $P2rx7^{-/-}$ mice 1 day before intranasal delivery of BLM or when indicated in repeated administration. Mice were treated i.p. every day for 14 days with vehicle (PBS, 10% DMSO) or with HEI3090 (1.5 mg/kg in PBS, 10% DMSO). $Nlrp3^{-/-}$ spleens were a kind gift from Dr Laurent Boyer, $Il18^{-/-}$ spleens from Dr George Birchenough and $Il1b^{-/-}$ spleens from Dr Bernhard Ryffel, all on a C57BL/6J background.

## Histology

Lungs were collected and fixed in 3% formamide for 16 hr prior inclusion in paraffin. Deparaffinized lungs sections (3 µm) were rehydrated and sequentially incubated with Hematoxylin, Mayer's, Bluing Reagent and Eosin Y Solution as indicated by the manufacturer (H&E staining, Abcam, ab245880) or Picro Sirius Red Solution followed by Acetic Acid as indicated by the manufacturer (Picro Sirius Red staining, Abcam, ab150681). The severity of fibrosis was assessed on whole lungs using the Ashcroft modified method which assigns eight grades (0: normal lung, 1: isolated alveolar septa with gentle fibrotic changes, 2: fibrotic changes of alveolar septa with knot-like formation, 3: contiguous fibrotic walls of alveolar septa, 4: single fibrotic masses, 5: confluent fibrotic masses, 6: large contiguous fibrotic masses, 7: air bubbles, and 8: fibrous obliteration to quantify lung fibrosis) to reliable and reproducible quantify fibrosis-induced lung remodeling (*Hübner et al., 2008*). To be even more accurate and not biased by patchy lesions observed in all existing lung fibrosis-induced mouse models, the whole lungs (left and right lobes) were divided in section of 880 µm² and each section was scored individually as shown in *Figure 2—figure supplement 1B*. A total of 80–110 sections were analyzed per mouse.

Levels of collagen on whole lungs were assessed on Sirius Red polarized light images of the entire lung taken with HD – Axio Observer Z1 Microscope ZEISS microscope (*Figure 2*) or with the SLIDEVIEW V200 slide scanner, Evident Europe Rungis (*Figure 4*). The collagen amount given by the polarization intensity of the Sirius Red staining of the lung slices and the percentage of fibrotic area was quantified with an ImageJ/Fiji macro program as following: The mean gray value and the integrated intensity of the collagen staining were measured in the fibrotic and non-fibrotic regions excluding the signal coming from vessels and bronchial tubes using dedicated masks. The binary masks were obtained after a median filtering and a manual thresholding, from the brightfield images for the fibrotic and non-fibrotic ones and the polarization images for the vessels and bronchial tubes. Vessels and bronchial tubes are excluded from fibrotic and non-fibrotic masks by intersecting each of those masks with the vessel and bronchial tubes one. The resulting masks are then also intersected with the polarization image to measure specifically the mean gray value and integrated intensity of fibrotic and non-fibrotic areas. The percentage of fibrotic area is obtained from the fibrotic and non-fibrotic masks without vessels and bronchial tubes as the ratio between the fibrotic area and the whole lung area without vessels and bronchial tubes (non-fibrotic + fibrotic).

## Cells phenotyping by flow cytometry

Lungs or spleens were collected and digested with the lung or spleen dissociation kit (Miltenyi Biotech, ref 130-095-927 and 130-095-928) according to the supplier's instructions. Red blood cells were lysed using ACK lysis buffer (Gibco, A1049201). Fc receptors were blocked using anti-CD16/32

(2.4G2) antibodies followed by surface staining by incubating cells on ice, for 20 min, with saturating concentrations of labeled Abs (Key resources table) in PBS, 5% fetal bovine serum (FBS), and 0.5% Ethylenediaminetetraacetic acid (EDTA). Tregs were identified using the transcription factor staining Buffer Set (eBioscience) for FoxP3 staining. Intracellular staining was performed after stimulation of single-cell suspensions with Phorbol 12-myristate 13-acetate (at 50 ng ml$^{-1}$, Sigma), ionomycin (0.5 µg ml$^{-1}$, Sigma), and 1 µl ml$^{-1}$ Golgi Plug (BD Biosciences) for 4 hr at 37°C 5% $CO_2$. Cells were then incubated with Live/Dead stain (Invitrogen), according to the manufacturer's protocol prior to surface staining. Fc receptors were blocked using anti-CD16/32 (2.4G2) antibodies followed by surface staining by incubating cells on ice, for 20 min, with saturating concentrations of labeled Abs (Key resource table) in PBS, 5% FBS and 0.5% EDTA. Cytofix/Cytoperm kit (BD biosciences) was used for intracellular staining by following the manufacturer's instructions. Samples were acquired on Cyto-FLEX LX (Beckman Coulter) and analyzed using FlowJo (LLC).

## ELISA

Sera of mice were collected at the end of the experiment and stored at −80°C before cytokine detection by ELISA using mouse IL-1 beta/IL-1F2 (R&D, DY401) and IL-18 (MBL, ref 7625) according to the supplier's instructions.

## Western blot

Single-cell suspensions of whole lungs were lysed with Laemmli buffer (10% glycerol, 3% sodium dodecyl sulfate [SDS], 10 mM $Na_2HPO_4$) with protease inhibitor cocktail (Roche). Proteins were separated on a 12% SDS–polyacrylamide gel electrophoresis gel and electro transferred onto Polyvinylidene Fluoride (PVDF) membranes, which were blocked for 30 min at room temperature (RT) with 3% bovine serum albumin or 5% milk. Membranes were incubated with primary antibodies (see Key resources table) diluted at 4°C overnight. Secondary antibodies (Promega) were incubated for 1 hr at RT. Immunoblot detection was achieved by exposure with a chemiluminescence imaging system (PXI Syngene, Ozyme) after membrane incubation with enhanced chemio luminescence(ECL; Immobilon Western, Millipore). The bands intensity values were normalized to that of β-actin using ImageJ software.

## Statistical analyses

Quantitative data were described and presented graphically as medians and interquartiles or means and standard deviations. The distribution normality was tested with the Shapiro's test and homoscedasticity with a Bartlett's test. For two categories, statistical comparisons were performed using the Student's $t$-test or the Mann–Whitney's test. For three and more categories, analysis of variance or non-parametric data with Kruskal–Wallis was performed to test variables expressed as categories versus continuous variables. If this test was significant, we used the Tukey's test to compare these categories and the Bonferroni's test to adjust the significant threshold. For the gene set enrichment analyses, bilateral Kolmogorov–Smirnov test, and false discovery rate were used.

All statistical analyses were performed by biostatistician using Prism8 program from GraphPad software. Tests of significance were two-tailed and considered significant with an alpha level of $p < 0.05$ (graphically: * for $p < 0.05$, ** for $p < 0.01$, *** for $p < 0.001$).

## Acknowledgements

The authors wish to thank Dr George Birchenough, Dr Laurent Boyer, and Dr Bernhard Ryffel for sharing Il18$^{-/-}$, Nlrp3$^{-/-}$, and Il1b$^{-/-}$ spleens, respectively. The authors acknowledge the PICMI faciliy from IRCAN Institute for Research on Cancer and Aging and microscopy facility from the Institut de Pharmacologie Moléculaire et Cellulaire part of the « Microscopie Imagerie Cytométrie d'Azur » GIS IBiSA labeled platform. We are grateful to Anna Zelena, EVIDENT Technology Center Europe GmbH, and Fabien Bertholle, EVIDENT Europe GmbH – French Branch, who gave us access to their slide scanner. We acknowledge Clement Molina for his help with the in vivo experiments during the revision of the paper. This publication is based upon discussion from PRESTO COST action CA21130 supported by COST (European Cooperation in Science and Technology). Funding: Ligue Nationale Contre le Cancer (SJH), Fondation pour la recherche médicale grant number #FDT202106013099 (SJH), ARC grant number ARCTHEM2021020003478 (SJH, VV-C), Cancéropôle PACA (VV-C) The French Government

National Research Agency, ANR through the 'Investments for the Future': program reference #ANR-11-LABX-0028-01 (PH), Executive Unit for Financing Higher Education, Research, Development and Innovation (UEFISCDI), Bucharest, Romania (grant number PN-III-P4-ID-PCE-2020-0818, acronym REPAIR) (AG).

# Additional information

## Competing interests

Alina Ghinet, Valerie Vouret-Craviari: has a patent related to HEI3090 25 (PCT/EP2019/058013). The other authors declare that no competing interests exist.

## Funding

| Funder | Grant reference number | Author |
|---|---|---|
| Fondation pour la Recherche Médicale | FDT202106013099 | Serena Janho dit Hreich |
| Association recherche pour le cancer | ARCTHEM2021020003478 | Serena Janho dit Hreich |
| National Research Agency | ANR-11-LABX-0028-01 | Paul Hofman |
| Ligue Contre le Cancer | | Valerie Vouret-Craviari |

The funders had no role in study design, data collection, and interpretation, or the decision to submit the work for publication.

## Author contributions

Serena Janho dit Hreich, Conceptualization, Data curation, Formal analysis, Validation, Investigation, Methodology, Writing – original draft; Thierry Juhel, Investigation; Sylvie Leroy, Veronique Hofman, Supervision, Validation, Methodology; Alina Ghinet, Methodology; Frederic Brau, Resources, Validation, Methodology; Paul Hofman, Validation, Visualization; Valerie Vouret-Craviari, Conceptualization, Formal analysis, Funding acquisition, Validation, Investigation, Methodology, Writing – original draft, Project administration, Writing – review and editing

## Author ORCIDs

Serena Janho dit Hreich (iD) http://orcid.org/0000-0002-4274-1419
Frederic Brau (iD) http://orcid.org/0000-0001-5967-5895
Valerie Vouret-Craviari (iD) https://orcid.org/0000-0003-4096-5926

## Ethics

All mouse studies were approved by the committee for Research and Ethics of the local authorities (CIEPAL #598, protocol number APAFIS 21052-2019060610506376) and followed the European directive 2010/63/UE, in agreement with the ARRIVE guidelines. Experiments were performed in accord with animal protection representative at IRCAN.

Reviewer #1 (Public Review): https://doi.org/10.7554/eLife.88138.4.sa1
Reviewer #2 (Public Review): https://doi.org/10.7554/eLife.88138.4.sa2
Author Response https://doi.org/10.7554/eLife.88138.4.sa3

# Additional files

## Supplementary files

• Transparent reporting form

## Data availability

All data are available in the main text or the supplementary materials. RNAseq data from IPF and control patients were retrieved from GEO database under the accession numbers (GSE47460, and GPL14550).

The following previously published datasets were used:

| Author(s) | Year | Dataset title | Dataset URL | Database and Identifier |
|---|---|---|---|---|
| Tedrow J | 2013 | Gene expression profiling of chronic lung disease for the Lung Genomics Research Consortium | https://www.ncbi.nlm.nih.gov/geo/query/acc.cgi?acc=GSE47460 | NCBI Gene Expression Omnibus, GSE47460 |
| Agilent Technologies | 2011 | Agilent-028004 SurePrint G3 Human GE 8x60K Microarray (Probe Name Version) | https://www.ncbi.nlm.nih.gov/geo/query/acc.cgi?acc=GPL14550 | NCBI Gene Expression Omnibus, GPL14550 |

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
