## [Editor Report · eLife assessment]

This study presents a potentially **valuable** discovery which indicates that activation of the P2RX7 pathway by the small molecule HEI3090 can reduce lung fibrosis after its establishment by inflammatory damage. If confirmed, the study could clarify the role of specific immune networks in the establishment and progression of lung fibrosis. The presented data and analyses showing the efficacy of HEI3090 small molecule acting via the P2RX7 pathway in reducing lung fibrosis are **solid**. The studies also show that genetic deletion of P2RX7 itself can reduce the extent of fibrosis. P2RX7 can thus have distinct effects in various phases of the development of lung fibrosis. There is a need for additional definitive studies that specifically identify the discrete phases of when inflammasome activation via P2RX7 signaling can worsen fibrosis versus when the same signaling can be beneficial. It also needs to be established whether distinct immune cell populations mediate the detrimental and beneficial effects of P2RX7 activation in lung fibrosis.

---

## [Referee Report · Reviewer #1 (Public Review)]

In this revised preprint the authors investigate whether a presumably allosteric P2RX7 activating compound that they previously discovered reduces fibrosis in a bleomycin mouse model. They chose this particular model as publicly available mRNA data indicate that the P2RX7 pathway is downregulated in idiopathic pulmonary fibrosis patients compared to control individuals. In their revised manuscript, the authors use three proxies of lung damage, Ashcroft score, collagen fibers, and CD140a+ cells, to assess lung damage following the administration of bleomycin. These metrics are significantly reduced on HEI3090 treatment. Additional data implicate specific immune cell infiltrates and cytokines, namely inflammatory macrophages and damped release of IL-17A, as potential mechanistic links between their compound and reduced fibrosis. Finally, the researchers transplant splenocytes from WT, NLRP3-KO, and IL-18-KO mice into animals lacking the P2RX7 receptor to specifically ascertain how the transplanted splenocytes, which are WT for P2RX7 receptor, respond to HEI3090 (a P2RX7 agonist). Based on these results, the authors conclude that HEI3090 enhanced IL-18 production through the P2RX7-NLRP3 inflammasome axis to dampen fibrosis.

These findings could be interesting to the field, as there are conflicting results as to whether NLRP3 activation contributes to fibrosis and if so, at what stage(s) (e.g., acute damage phase versus progression). The revised manuscript is more convincing in that three orthogonal metrics for lung damage were quantified.

However, deletion of the P2RX7 receptor itself reduces the extent of fibrosis, suggesting that P2RX7 signaling can be pro-fibrotic. In the absence of P2RX7, the effects of HEI3900 are also abolished, suggesting that HEI3900 acts in part via P2RX7 signaling. This suggests a paradox that P2RX7 signaling can be both detrimental and beneficial in fibrosis and there is need for a better understanding of when P2RX7 signaling is beneficial and when it is detrimental in lung fibrosis. HEI3900-induced activation of P2RX7 seems to be beneficial but this primarily is shown for when fibrosis is already established. As the P2RX7 genetic deletion mouse model has less fibrosis, P2RX7 signaling and inflammasome activation may be deleterious during the formation of disease but it is also possible that HEI3900 has other beneficial effects that are not directly related to P2RX7.

Molecularly, additional evidence on specificity, such as thermal proteome profiling and direct biophysical binding experiments, would also enhance the authors' argument that the compound indeed binds P2RX7 directly and specifically. Since all small molecules have some degree of promiscuity, the absence of an additional P2RX7 modulator, or direct recombinant IL-18 administration, is needed to orthogonally validate the functional importance of this pathway. Another way the authors could probe pathway specificity would involve co-administering α-IL-18 with HEI3090 in several key experiments (similar to Figure 4L).

---

## [Referee Report · Reviewer #2 (Public Review)]

In the study by Hreich et al, the potency of P2RX7-specific positive modulator HEI3090, developed by the authors, for the treatment of Idiopathic pulmonary fibrosis (IPF) was investigated. Recently, the authors have shown that HEI3090 can protect against lung cancer by stimulating dendritic cell P2RX7, resulting in IL-18 production that stimulates IFN-γ production by T and NK cells (DOI: 10.1038/s41467-021-20912-2). Interestingly, HEI3090 increases IL-18 levels only in the presence of high eATP. Since the treatment options for IPF are limited, new therapeutic strategies and targets are needed. The authors first show that P2RX7/IL-18/IFNG axis is downregulated in patients with IPF. Next, they used a bleomycin-induced lung fibrosis mouse model to show that the use of a positive modulator of P2RX7 leads to the activation of the P2RX7/IL-18 axis in immune cells that limits lung fibrosis onset or progression. Mechanistically, treatment with HEI3090 enhanced IL-18-dependent IFN-γ production by lung T cells leading to a decreased production of IL-17 and TGFβ, major drivers of IPF. The major novelty is the use of the small molecule HEI3090 to stimulate the immune system to limit lung fibrosis progression by targeting the P2RX7, which could be potentially combined with current therapies available. Overall, the study was well performed and the manuscript is clear.

---

## [Author Response]

The following is the authors’ response to the previous reviews.

**Point to point response for the editors**

We are deeply grateful for the time you have devoted to reviewing this manuscript, and we sincerely thank you. Your insightful feedback has been instrumental in enhancing the quality of our work.

In the revised version of the manuscript, we have carefully addressed each of the concerns you raised. Below, you will find a detailed summary of how your feedback has been incorporated to improve the overall content and clarity of the document.

1. P2RX7 effects: In Figure 2, the vehicle treated P2RX7 knockout (panel M) shows an Ashcroft score of about 1.5 after BLM. Comparing this to the Ashcroft score of 3 after BLM in the wildtype (panel C) suggests that P2RX7 deletion is an effective way to reduce fibrosis by half!.The argument that HEI3090 also reduces fibrosis by activating P2RX7 is of course very difficult to convey and it seems contradictory that P2RX7 deletion and P2RX7 activation can be both anti-fibrotic. This is an unusual claim and confuses the reviewers as well as the future readers.This has many important health implications because activating an inflammatory pathway via P2RX7 and IL-18 could be risky in terms of a fibrosis treatment as inflammatory activation can also worsen fibrosis. The authors' own P2RX7 KO data (untreated vehicle groups) indeed confirms that P2RX7 can be pro-fibrotic.

We thank the editors for their comment highlighting the lack of clarity in our message. Indeed, we verified whether the antifibrotic action of HEI3090 depends on the expression of P2RX7 by inducing lung fibrosis in P2RX7 KO mice. In doing so, we initially observed that P2RX7 plays a role in the development of BLM-induced lung fibrosis. This is illustrated by a decrease of 50% in the Ashcroft score, as shown in Figure 2M and Supplemental Figure 2C of the revised manuscript.

To increase the clarity of your message, we added in the text the following paragraph:

"We further verified whether the antifibrotic action of HEI3090 depends on the expression of P2RX7 by inducing lung fibrosis in p2rx7 knockout (KO) mice. In doing so, we initially observed that P2RX7 plays a role in the development of BLM-induced lung fibrosis. This is illustrated by a decrease of 50% in the Ashcroft score, with a mean value of 1.7 in P2RX7 knockout mice compared to 3 in wild-type mice (Figure 2M and Supplemental Figure 2C). It is important to note that p2rx7 -/- mice still exhibit signs of lung fibrosis, such as thickening of the alveolar wall and a reduction in free air space, in comparison to naïve mice that received PBS instead of BLM (see Supplemental Figure 2A). This result confirms a previous report indicating that BLM-induced lung fibrosis partially depends on the activation of the P2RX7/pannexin-1 axis, leading to the production of IL-1β in the lung. Additionally, in contrast to the observations in WT mice, HEI3090 failed to attenuate the remaining lung fibrosis in p2rx7 -/- mice, as measured by the Ashcroft score (Figure 2M), the percentage of lung tissue with fibrotic lesions, or the intensity of collagen fibers (Supplemental Figure 2D). These results show that P2RX7 alone participates in fibrosis and that HEI3090 exerts a specific antifibrotic effect through this receptor (see Supplemental Figure 2C)."

Since we used the HEI3090 compound in this study and to be closer to the results, we have replaced the title of 2 chapters in the results section as followed:

“HEI3090 inhibits the onset of pulmonary fibrosis in the bleomycin mouse model” instead of P2RX7 activation inhibits the onset of pulmonary fibrosis in the bleomycin mouse model and“HEI3090 shapes immune cell infiltration in the lungs" instead of P2RX7 activation shapes immune cell infiltration in the lungs

We concur that the observation of both anti-fibrotic effects following P2RX7 deletion and P2RX7 activation appears contradictory. This specific aspect has been thoroughly addressed and extensively discussed in the revised manuscript.

“A major unmet need in the field of IPF is new treatment to fight this uncurable disease. In this preclinical study, we demonstrate the ability of immune cells to limit lung fibrosis progression. Based on the hypothesis that a local activation of a T cell immune response and upregulation of IFN-γ production has antifibrotic proprieties, we used the HEI3090 positive modulator of the purinergic receptor P2RX7, previously developed in our laboratory (Douguet et al., 2021), to demonstrate that activation of the P2RX7/IL-18 pathway attenuates lung fibrosis in the bleomycin mouse model. We have demonstrated that lung fibrosis progression is inhibited by HEI3090 in the fibrotic phase but also in the acute phase of the BLM fibrosis mouse model, i.e. during the period of inflammation. This lung fibrosis mouse model commonly employed in preclinical investigations, has recently been recognized as the optimal model for studying IPF (Jenkins et al., 2017). In this model, the intrapulmonary administration of BLM induces DNA damage in alveolar epithelial type 1 cells, triggering cellular demise and the release of ATP. The extracellular release of ATP from injured cells activates the P2RX7/pannexin 1 axis, initiating the maturation of IL1β and subsequent induction of inflammation and fibrosis. In line with this, mice lacking P2RX7 exhibited reduced neutrophil counts in their bronchoalveolar fluids and decreased levels of IL1β in their lungs compared to WT mice (Riteau et al., 2010). Based on these findings, Riteau and colleagues postulated that the inhibition of P2RX7 activity may offer a potential strategy for the therapeutic control of fibrosis in lung injury. In the present study we provided strong evidence showing that selective activation of P2RX7 on immune cells, through the use of HEI3090, can dampen inflammation and fibrosis by releasing IL-18. The efficacy of HEI3090 to inhibit lung fibrosis was evaluated histologically on the whole lung’s surface by evaluating the severity of fibrosis using three independent approaches applied to the whole lung, the Ashcroft score, quantification of fibroblasts/myofibroblasts (CD140a) and polarized-light microscopy of Sirius Red staining to quantify collagen fibers. All these methods of fibrosis assessment revealed that HEI3090 exerts an inhibitory effect on lung fibrosis, underscoring the necessity for a thorough pre-clinical assessment of HEI3090's mode of action. Notably, HEI3090 functions as an activator, rather than an inhibitor, of P2RX7, further emphasizing the importance of elucidating its intricate mechanisms.”

We trust that the detailed explanation provided therein will adequately persuade both the reviewers and future readers.

1. The statistical concerns are based on the phrasing of "the experiment was stopped when significantly statistical results were observed". This is different from the power analysis approach that the authors describe in their latest rebuttal. However, it raises the question why the power analysis was performed using "on a one-way ANOVA analysis comparing in each experiment the vehicle and the treated group". The analyses in the manuscript use the Mann-Whitney test for several comparisons which ahs the assumption that the samples do NOT have a normal distribution. An ANOVA and t-tests have the assumption that samples are normally distributed. If the power analysis and "statistical forecasting" assumed a normal distribution and used an ANOVA, then shouldn't all the analyses also use a statistical test appropriate for normally distributed samples such as ANOVA and t-tests?Several of the data points in the figures seem to be normally distributed and therefore t-test for two group comparisons would be more appropriate. The most rigorous approach would be to check for normal distribution before choosing the correct statistical test and using the t-test/ANOVA in normally distributed data as well as Mann-Whitney for non-normally distributed data.

We described in the Material and Method section of the revised manuscript our approach to determine the size of experimental group.

“The determination of experimental group sizes involved conducting a pilot experiment with four mice in each group. Subsequently, a power analysis, based on the pilot experiment's findings (which revealed a 40% difference with a standard error of 0.9, α risk of 0.05, and power of 0.8), was performed to ascertain the appropriate group size for studying the effects of HEI3090 on BLM-induced lung fibrosis. The results of the pilot experiment and power analysis indicated that a group size of four mice was sufficient to characterize the observed effects. For each full-scale experiment, we initiated the study with 6 to 8 mice per group, ensuring a minimum of 5 mice in each group for robust statistical analysis. Additionally, we systematically employed the ROULT method to identify and subsequently exclude any outliers present in each experiment before conducting statistical analyses”.

We now described in the Material and Method section how we carried out the statistical analyses.

“Quantitative data were described and presented graphically as medians and interquartiles or means and standard deviations. The distribution normality was tested with the Shapiro's test and homoscedasticity with a Bartlett's test. For two categories, statistical comparisons were performed using the Student's t-test or the Mann–Whitney's test. For three and more categories, analysis of variance (ANOVA) or non-parametric data with Kruskal–Wallis was performed to test variables expressed as categories versus continuous variables. If this test was significant, we used the Tukey's test to compare these categories and the Bonferroni’s test to adjust the significant threshold. For the Gene Set Enrichment Analyses (GSEA), bilateral Kolmogorov–Smirnov test, and false discovery rate (FDR) were used.All statistical analyses were performed by biostatistician using Prism8 program from GraphPad software. Tests of significance was two-tailed and considered significant with an alpha level of P < 0.05. (graphically: * for P < 0.05, ** for P < 0.01, *** for P < 0.001).”

We also added in the legend of each figure, the statistical analysis used to determine each p-values.

1. Adoptive transfer: The concerns of the reviewers include an unclear analysis of the effects of adoptive transfer itself and the approaches used to analyze the data independent of the HEI3090 effect. For example, in Figure 4, the adoptive transfer IL18-/- cells (vehicle group) leads to an Ashcroft score of about 1 and among the lowest of the BLM exposed mice. Does that mean that IL18 is pro-fibrotic and that its absence is beneficial? If yes, it would go against the core premise of the study that IL18 is beneficial. Statistical comparisons of the all the vehicle conditions in the adoptive transfer would help clarify whether adoptive transfer of NLRP3-/-, IL18-/- in wild-type and P2RX7-/- mice reduces or increases fibrosis. Such multiple comparisons are necessary to fully understand the adoptive transfer studies and would also require the appropriate statistical test with corrections for multiple comparisons such as Kruskal-Wallis for data without normal distribution and ANOVA with post hoc correction for normal distribution.

We added a new paragraph in the revised version of the manuscript to explain the adoptive transfer approach.

“We wanted to further investigate the mechanism of action of HEI3090 by identifying the cellular compartment and signaling pathway required for its activity. Since the expression of P2RX7 and the P2RX7-dependent release of IL-18 are mostly associated with immune cells (Ferrari et al., 2006), and since HEI3090 shapes the lung immune landscape (Figure 3), we investigated whether immune cells were required for the antifibrotic effect of HEI3090. To do so, we conducted adoptive transfer experiments wherein immune cells from a donor mouse were intravenously injected one day before BLM administration into an acceptor mouse. The intravenous injection route was chosen as it is a standard method for targeting the lungs, as previously documented (Wei and Zhao, 2014). This approach was previously used with success in our laboratory (Douguet et al., 2021). It is noteworthy that this adoptive transfer approach did not influence the response to HEI3090. This was observed consistently in both p2rx7 -/- mice and p2rx7 -/- mice that received splenocytes of the same genetic background. In both cases, HEI3090 failed to mitigate lung fibrosis, as depicted in Figure 2M and Supplemental Figures 2D and 6A and B.”

We added the Supplemental Figure 7 showing that the genetic background does not impact lung fibrosis at steady step levels where p-values were analyzed by one-way ANOVA, with Kruskal-Wallis test for multiple comparisons.

**Author response image 1. sa3fig1:** Supplemental Figure 7 : The genetic background does not impact lung fibrosis at steady step levels. p2rx7-/- mice were given 3.106 WT, nlrp3-/ , i118-/ or illb -l- splenocytes i_v_ one day prior to BLM delivery (i_n_ 2.5 LJ/kg)_ p2rx7-/- mice or p2rx7-/- mice adoptively transferred with splenocytes from indicated genetic background were treated daily i.p_ with mg/kg HE13090 or vehicle for 14 days. Fibrosis score assessed by the Ashcroft method. P-values were analyzed on all treated and non treated groups by one-way ANOVA, with Kruskal-Wallis test for multiple comparisons. The violin plot illustrates the distribution of Ashcroft scores across indicated experimental groups. The width of the violin at each point represents the density of data, and the central line indicates the median expression level. Each point represents one biological replicate. ns, not significant